# Proteomics signature of moderate-to-vigorous physical activity and risk of multimorbidity of cancer and cardiometabolic diseases

Michael J. Stein [1]✉, Hansjörg Baurecht [1,2], Patricia Bohmann[1], Reynalda Cordova[3], Pietro Ferrari[4], Béatrice Fervers[5,6], Christine M. Friedenreich [7,8], Marc J. Gunter[9], Laia Peruchet-Noray [4], Diana Wu [4], Charlotte Onland-Moret [10], Maria-José Sánchez[11,12,13], María-Dolores Chirlaque[13,14], Michael F. Leitzmann[1], Vivian Viallon[4] & Heinz Freisling [4]

## Abstract

**Background** Moderate-to-vigorous physical activity (MVPA) is inversely associated with risks of cancer, cardiovascular diseases (CVD), type 2 diabetes (T2D), and their co-occurrence, defined as multimorbidity; however, the underlying biological pathways remain unclear.
**Methods** In 33,806 UK Biobank participants with 2911 measured blood proteins, a proteomic signature of MVPA was derived with linear and LASSO regressions. Multivariable Cox models, adjusted for MVPA, estimated prospective associations with cancer, CVD, T2D, and multimorbidity.
**Results** We show that after multiple testing corrections, 220 proteins are retained in the MVPA signature. Proteins related to food intake, metabolism, and cell growth (e.g., LEP, MSTN) are inversely associated, while those involved in immune cell migration and musculoskeletal integrity (e.g., integrins, COMP) are positively associated with MVPA. Several proteins positively associated with MVPA are inversely associated with disease risk (e.g., integrins, CLEC4A for cancer; LPL, LEP for T2D), while proteins negatively associated with MVPA are positively associated with disease risk (e.g., CD38, TGFA for CVD). The proteomic signature score is inversely associated with cancer risk (hazard ratio per interquartile range: 0.87; 95% confidence interval: 0.78, 0.96) and T2D (0.66; 0.60, 0.72). For multimorbidity, proteins inversely related to MVPA align with expected risk patterns (e.g., GGT1, HR: 1.32; 95% CI: 1.12, 1.57), but the proteomic signature score is not associated.
**Conclusions** This study identifies several proteins associated with MVPA that are also associated with cancer, CVD, T2D, and the multimorbidity of these conditions. Further studies investigating the causal nature of these associations are welcome.

## Plain language summary

Many people develop more than one long-term disease, such as cancer, heart disease, or diabetes, a condition known as multi-morbidity. While regular physical activity is known to reduce the risk of these diseases, the underlying biological pathways remain poorly understood. In this study, we analyzed thousands of proteins in the blood of over 33,000 adults participating in UK Biobank, a large health study, to identify proteins linked with higher levels of physical activity. We discovered several proteins that were either more or less common in active individuals – and these same proteins were also associated with a lower risk of developing cancer, heart disease, diabetes, and multimorbidity. These findings suggest that physically activity may help prevent multiple diseases by influencing proteins in the blood. This opens new possibilities for research into how physical activity supports long-term health.

Cancer, cardiovascular diseases (CVD), and type 2 diabetes (T2D) are leading causes of death globally[1]. The global burden of each of these diseases approximately doubled from 1990 to 2019[2–4]. Four out of ten individuals with cancer have at least one other chronic disease, most commonly CVD or T2D[5]. This co-occurrence of multiple conditions in an individual is referred to as multimorbidity[6] and may be explained by shared risk factors across diseases[7]. While physical activity is convincingly related to decreased risk of cancer, CVD, and T2D[8], its associations with multimorbidity remain less clear. Studies employing a broad definition of multimorbidity – encompassing a wide range of diseases – have reported inverse associations[9],

whereas studies focusing on more specific disease combinations have found no association[10,11]. Given the heterogeneity of multimorbidity, broad definitions may limit mechanistic interpretability. Focusing on commonly co-occurring diseases with shared etiological pathways may provide more informative insights[12]. Although cancer, T2D, and CVD are inherently heterogeneous with organ-specific pathophysiology, they share molecular mechanisms ("hallmarks"), particularly chronic inflammation, which may contribute to multimorbidity.

The biological pathways through which physical activity, particularly at higher intensities[13], may reduce the risk of these diseases involve improved insulin sensitivity, reduced chronic inflammation, and modulation of sex hormone activity[14]. However, biomarkers associated with these pathways cover only a subset of potential mechanisms and are often insensitive to physical activity interventions.

Physical activity is a multidimensional exposure, encompassing frequency, intensity, duration, and type – each of which influences distinct physiological systems[15]. Proteomics offers a tool to capture these dimensions, because circulating proteins reflect underlying biological pathways and physiological processes[16] that are activated by different aspects of habitual physical activity. Most circulating proteins are also relatively stable over months to years[17], making them reliable markers for studying long-term adaptations to habitual physical activity[18–20].

Three previous cohort studies have analyzed physical activity in relation to the proteome. A smaller study ($n = 897$) identified five proteins that were associated with physical activity[21], while a larger study ($n = 11,695$) found 65 proteins associated with physical activity[22]. In 2025, a UK Biobank study ($n = 39,160$) identified 41 proteins consistently associated with various physical activity measures, including self-reported and device-based physical activity, and associated these proteins with dementia risk[23]. These studies highlight the potential to identify physical activity-related biochemical pathways with health relevance. However, associations with cancer and cardiometabolic diseases remain unexplored.

In this study, we assess large-scale proteomic data to derive a proteomic signature of total moderate-to-vigorous physical activity (MVPA) and investigate its association with cancer, CVD, T2D, and multimorbidity. This signature captures protein patterns reflecting systemic pathways related to inflammation, metabolism, and tissue integrity. MVPA-associated proteins are associated with a lower risk of incident cancer, CVD, T2D, and the development of multimorbidity. These findings provide insights into multiple pathways associated with habitual physical activity and chronic disease risk.

## Methods
### Study design and participants
UK Biobank is a prospective cohort study that enrolled 502,134 UK participants aged 40 to 69 years between 2006 and 2010. The study collected sociodemographic, lifestyle, and extensive phenotypic data using touchscreen questionnaires, interviews, physical and functional measurements, and biomaterial collection. Ethics approval was obtained from the North West Multi-Center Research Ethics Committee, and all participants provided written informed consent[24]. The UK Biobank Pharma Proteomics Project (UKB-PPP) is a precompetitive consortium that generated protein measurements using the Olink Proximity Extension Assay, capturing 2923 unique proteins in 53,013 participants. Details of the inclusion process for the UKB-PPP are described elsewhere[25]. In brief, an initial sample of 6229 participants was pre-selected by UKB-PPP members, and the remaining samples were selected from the UK Biobank, ensuring representation of the full cohort in terms of age, sex, and recruitment center.

After exclusion of prevalent cases of cancer, CVD, and T2D as well as incident cases during the first year of follow-up ($n = 8743$), participants with missing information on MVPA or covariate data ($n = 8409$), and those with >50% missingness in protein data ($n = 2055$), a total of 33,806 participants were included in the analyses (see Supplementary Fig. 1 for a flowchart of participant exclusions).

### Physical activity measurement
MVPA was assessed using the International Physical Activity Questionnaire (IPAQ) Short Form[26], which captures weekly frequency and daily duration (minimum 10 minutes) of walking, moderate, and vigorous physical activity over the previous four weeks. The IPAQ Short Form demonstrated acceptable reliability and validity against actigraphy[27]. Responses may still be affected by recall and social desirability bias. Based on evidence that random error dominates in large-scale physical activity measurements, associations of physical activity with circulating proteins, and disease outcomes are most likely underestimated[28]. Following the IPAQ evaluation protocol[26], Metabolic Equivalent of Task (MET) values from the Compendium of Physical Activities developed by Ainsworth and colleagues[29] for moderate (4.0 METs) and vigorous (8.0 METs) activities were multiplied by the respective frequency and duration to estimate total MET-hours per week (MET-hours/week) of MVPA. No distinction was made between recreational and occupational activities. We focused on MVPA due to its purportedly stronger physiological effects compared to activities of light intensity. MVPA was rescaled to have a mean of 0 and a standard deviation (SD) of 1.

### Protein measurements and proteomic data processing
The Olink assay technology and the analyses of the UKB-PPP are detailed elsewhere[25,30]. In brief, the relative abundance of 2923 proteins was quantified using antibodies distributed across four 384-plex panels: inflammation, oncology, cardiometabolic, and neurology. Blood samples were assayed in four 384-well plates consisting of four abundance blocks for each of the four panels per 96 samples, using the Olink Explore platform. This platform is based on proximity extension assays, which are highly sensitive and reproducible, with low cross-reactivity. Relative concentrations of the 2923 unique proteins were measured by next-generation sequencing and expressed as normalized protein expression (NPX) values on a log-base-2 scale. Following a previously published approach[31], protein values below the limit of detection (LOD) were replaced with the LOD divided by the square root of two[32]. Proteins with more than 20% missingness (n = 12) were excluded. Missing protein expression levels were imputed using k-nearest neighbors ($k = 10$), allowing for up to 50% missingness across proteins and 20% per protein. Finally, each protein was rescaled to have a mean of 0 and a standard deviation (SD) of 1.

### Cohort follow-up and outcome ascertainment
Participants' vital status was obtained through linkage with health care data and national death registries[33]. Follow-up began at baseline and ended at the date of complete follow-up (December 2016 for Wales, December 2020 for England, November 2021 for Scotland)[34], loss to follow-up, or death, whichever came first. Follow-up for incident diseases and subsequent multimorbidity was structured using a Lexis object, which allows participants' follow-up time to be split into intervals corresponding to distinct risk periods[35]. For cancer incidence, we focused on the first primary malignant cancers with convincing evidence for an association with physical activity[36], including cancers of the breast, colorectum, corpus uteri, bladder, kidney (renal cell carcinoma), esophagus (adenocarcinoma), and stomach (cardia), which were combined into a composite "cancer" outcome. The composite CVD outcome included diagnoses of angina pectoris, myocardial infarction, ischemic heart diseases, arterial fibrillation, cardiac arrhythmias, heart failure, cerebrovascular diseases and stroke, atherosclerosis, and other peripheral vascular diseases. T2D was identified using the International Classification of Diseases (ICD)−10 code E11. A complete list of all ICD codes for all outcomes is provided in Supplementary Table 1.

Multimorbidity was defined as the presence of any combination of at least two of the following outcomes: cancer, CVD, or T2D. When two diseases occurred on the same day (n = 195), they were separated by one day based on an arbitrary temporal order (T2D, cancer, CVD). To address reverse causation and detection bias, we excluded the first year of follow-up after diagnoses of primary outcomes (175, 405, and 349 exclusions for multimorbidity following cancer, CVD, and T2D, respectively).

## Statistics and reproducibility

**Proteomic analyses of MVPA.** We identified proteins associated with MVPA using a two-step procedure. First, we used linear regression separately for each of the 2911 proteins (predictors) and MVPA as the outcome with false discovery rate-corrected *q*-values as the criterion. Each model also included sex, age, study center, body mass index, diet, alcohol, and smoking as covariates. Second, we applied the least absolute shrinkage and selection operator (LASSO) regression to these pre-filtered proteins and selected those with non-zero coefficients as the most predictive proteins of MVPA. Covariates, as mentioned above, were included but not penalized in the LASSO model. To select the optimal value of the regularization parameter, we performed LASSO regression with fivefold cross-validation for the assessment of the association with MVPA, tuning the regularization parameter lambda across 100 values from the maximum down to 0.1% of that value, minimizing the mean squared error.

LASSO-retained proteins were used to create an MVPA proteomics signature score for each participant, calculated as the weighted sum of protein levels using the respective ß-coefficients from the LASSO model as weights.

We used a fivefold nested cross-validation to assess the strength of association of the model with MVPA. While the model with the lowest prediction error yielded a slightly higher correlation (0.34, 95% confidence interval (CI): 0.33, 0.35; $R^2 = 0.12$), we chose a more conservative regularization level (prediction error within one standard error of the minimum) that resulted in fewer selected proteins. This model achieved a correlation of 0.32 (95% CI: 0.31, 0.33; $R^2 = 0.10$), indicating a moderate positive relationship between the proteomic signature and MVPA in out-of-sample prediction.

**Protein-protein interaction networks and pathway enrichment analyses.** To gain insights into the underlying biological mechanisms of the identified proteins, we constructed protein-protein interaction networks using STRING[37] and visualized them with Cytoscape[38]. To identify proteins with the highest number of interactions, the maximal clique centrality (MCC) method was utilized[39].

We conducted pathway enrichment analyses to investigate whether MVPA-associated proteins clustered within distinct biological pathways. Using the Gene Ontology resource[40], we examined enrichment across biological processes, molecular functions, and cellular components. We also assessed pathway-level enrichment using Kyoto Encyclopedia of Genes and Genomes (KEGG)[41], Reactome[42], and WikiPathways[43]. Enrichment tests were performed against a background gene set corresponding to all proteins included in the analysis.

**Associations with disease endpoints.** We used Cox proportional hazard regression with age as the underlying time metric[44] to estimate hazard ratios (HRs) and corresponding 95% CIs for the associations between individual MVPA-associated proteins, and separately, the proteomic signature score, with cancer, CVD, T2D, and multimorbidity. The Cox models were stratified by age at baseline (5-year increments), sex, and country (England, Scotland, Wales), and adjusted for education level (highest, intermediate, lowest, none of those), socio-economic status (Townsend index, categorized using tertiles), smoking (never, former, current), alcohol use (never, former, current), sedentary behavior (0–3 h, 4–5 h, 6–7 h, >8 h of daily TV watching, PC use during leisure, and driving), and screening for breast and/or bowel cancer (binary) as categorical variables, as well as MVPA (MET-hours), body mass index (kg/m$^2$), and diet (healthy diet score, 0–7 scale)[45] as continuous variables. Missing values were coded as missing. Non-linearity was addressed using restricted cubic splines with four knots at the 0.05, 0.35, 0.65, and 0.95 quantiles. Departures from linearity were evaluated by comparing models with and without spline terms using log-likelihood ratio tests. The proportional hazards assumption was tested using Schoenfeld residuals, with no evidence of violation observed.

In sensitivity analyses, we evaluated whether excluding MVPA-adjustment affected the association between the proteomic signature and disease risk by assessing model fit with likelihood ratio tests, Akaike Information Criterion (AIC), and the concordance index (Harrell's C-index). Second, we estimated HRs and 95% CIs with and without mutual adjustment to explore whether the proteomic signature (partly) explains the association between MVPA and all disease transitions. Third, we added ethnicity as a non-penalized covariate to the LASSO model to assess potential confounding. Fourth, we included participants with incident disease events during the first year of follow-up to evaluate potential bias from latent disease. Lastly, to ensure robustness of the identified physical activity-associated proteins, we performed a cross-platform validation in the European Prospective Investigation into Cancer and Nutrition (EPIC) cohort (1992–2000). EPIC was approved by the Ethical Review Boards of the International Agency for Research on Cancer and the Institutional Review Board of each participating center, and all participants provided written informed consent. After excluding participants with missing physical activity data, SomaScan 7k proteomic data were available for 17,273 individuals. About 1800 proteins are targeted by both Olink and SomaScan[46]. All *P*-values to assess exposure outcome associations and for enrichment analyses were adjusted for multiple comparisons using the Benjamini-Hochberg procedure (false discovery rate *q*-values) with a significance threshold of 5%.

All data processing and statistical analyses were performed using R 4.4[47]. Cox regression analyses were performed using the *rms* package[48].

## Results

We assessed proteome–disease associations in 33,806 participants (53% women, median age 57 years [interquartile range: 49 to 63]) (Table 1). During a median follow-up of 11.8 years (interquartile range: 11.1 to 12.5), 1108 participants developed cancer, 3445 developed CVD, 1363 developed T2D, and 814 developed multimorbidity (Fig. 1A, Supplementary Table 2).

### Physical activity-related proteins

After pre-filtering (870 proteins associated with MVPA) and subsequent LASSO regression, we identified 220 proteins with non-zero coefficients, of which 118 were inversely and 102 positively associated with MVPA. Supplementary Data 1 provides regression coefficients with the full names of all identified proteins. Pairwise correlations revealed weak to moderate correlations among these proteins (Fig. 1B). MVPA was strongly inversely associated with the proteins LEP, MSTN, and TGFBR2, and strongly positively associated with COMP, ITGAM, and MYOM3 (Fig. 1C).

### Protein-protein interaction networks and pathway enrichment analysis

Among the proteins that were inversely associated with MVPA, 78 proteins formed 119 protein-protein interactions, with LEP, ERBB2, and IGFBP3 emerging as the three hub proteins (Fig. 2). Among the proteins that were positively associated with MVPA, 75 proteins formed 196 protein-protein interactions with an interconnected network of integrin and the three hub proteins ITGAM, CDH5, and CD34.

Pathway enrichment analyses revealed that proteins inversely associated with MVPA were enriched in extracellular region components and multicellular organism processes. Proteins positively associated with MVPA were enriched in cell surface and membrane-associated complexes, particularly integrin complexes and receptor signaling structures, suggesting roles in cell adhesion and communication. Molecular functions included integrin and opsonin binding, indicating involvement in extracellular matrix interactions and immune recognition (Table 2).

### Proteome–disease associations

The proteome-disease associations of all 220 proteins and the proteomic signature score of MVPA are shown in a volcano plot for all disease transitions (Fig. 3).

## Table 1 | Baseline population characteristics of the UK Biobank proteomics cohort

| Characteristic | Overall | Women | Men |
|---|---|---|---|
| N | 33,806 | 17,898 | 15,908 |
| Physical activity (MET-hours) | 27.0 (33.5) | 24.8 (29.9) | 29.4 (37.1) |
| Age (years) | 55.8 (8.2) | 55.7 (8.1) | 56.0 (8.3) |
| Country | | | |
| England | 29,976 (88.7%) | 15,863 (88.6%) | 14,113 (88.7%) |
| Scotland | 2452 (7.3%) | 1312 (7.3%) | 1140 (7.2%) |
| Wales | 1378 (4.1%) | 723 (4.0%) | 655 (4.1%) |
| Body mass index (kg/m²) | 27.2 (4.6) | 26.8 (5.0) | 27.5 (4.0) |
| Townsend Index | −1.4 (3.1) | −1.4 (3.0) | −1.4 (3.1) |
| Missing | 42 (0.1%) | 15 (0.1%) | 27 (0.2%) |
| Education | | | |
| Highest | 12,394 (36.7%) | 6289 (35.1%) | 6105 (38.4%) |
| Intermediate | 7904 (23.4%) | 4083 (22.8%) | 3821 (24.0%) |
| Lowest | 8812 (26.1%) | 5052 (28.2%) | 3760 (23.6%) |
| Other | 4462 (13.2%) | 2357 (13.2%) | 2105 (13.2%) |
| Missing | 234 (0.7%) | 117 (0.7%) | 117 (0.7%) |
| Smoking status | | | |
| Never | 19,030 (56.3%) | 10,873 (60.7%) | 8157 (51.3%) |
| Former | 11,397 (33.7%) | 5528 (30.9%) | 5869 (36.9%) |
| Current | 3379 (10.0%) | 1497 (8.4%) | 1882 (11.8%) |
| Alcohol use | | | |
| Never | 1356 (4.0%) | 916 (5.1%) | 440 (2.8%) |
| Former | 1125 (3.3%) | 626 (3.5%) | 499 (3.1%) |
| Current | 31,325 (92.7%) | 16,356 (91.4%) | 14,969 (94.1%) |
| Healthy diet score | 3.6 (1.4) | 3.9 (1.2) | 3.3 (1.4) |
| Sedentary behavior (hours) | 4.5 (2.6) | 4.0 (2.3) | 5.0 (2.7) |
| Missing | 4 | 2 | 2 |
| Cancer screening | | | |
| Never | 14,714 (43.5%) | 3608 (20.2%) | 11,106 (69.8%) |
| Ever | 19,092 (56.5%) | 14,290 (79.8%) | 4802 (30.2%) |

*MET* metabolic equivalent of task.

Numbers show mean (standard deviations) or *N* (percentage).

For the transition to cancer, we identified 13 proteins associated with cancer risk after adjusting for covariates and MVPA, five of which were inversely related and seven positively related. Among the top hits, CHRDL2 (HR per interquartile range: 1.18; 95% CI: 1.10, 1.27, $q = 8.26 \times 10^{-5}$) and PI3 (HR per interquartile range: 1.14; 95% CI: 1.06, 1.22, $q = 0.006$) were strongly positively associated with cancer risk, while CLEC4A showed a non-linear inverse association (*P*-nonlinear = 0.007). The proteomics signature score was inversely associated with cancer risk (HR: 0.87; 95% CI: 0.79, 0.96, $q = 0.037$).

For the transition to CVD, 51 proteins were associated with a decreased risk, and 46 proteins with an increased risk. Among these, EDA2R emerged as the most significant hit (HR per interquartile range: 1.21; 95% CI: 1.09, 1.34, $q = 5.67 \times 10^{-24}$; P-nonlinear=0.016). The proteomics signature score was inversely associated with CVD risk (HR per interquartile range: 0.93;

95% CI: 0.85, 1.02; $q = 0.088$; P-nonlinear=0.014); not passing the false discovery rate threshold.

For the transition to T2D, 63 proteins were associated with a decreased risk, and 90 proteins with an increased risk. Among these, MXRA8 emerged as the most significant hit (HR per interquartile range: 0.51, 95% CI: 0.48, 0.55; $q = 1.30 \times 10^{-67}$). The proteomics signature score was inversely associated with T2D risk (HR per interquartile range: 0.67; 95% CI: 0.62, 0.74; $q = 1.19 \times 10^{-16}$).

Figure 4 highlights proteins significantly associated with at least one outcome (false discovery rate-corrected). Supplementary Fig. 2 shows linear and non-linear HR curves for signature scores and top hits per transition.

### Proteins related to multimorbidity

For multimorbidity following a cancer diagnosis, we identified two proteins that were inversely associated and three proteins that were positively associated with the risk of multimorbidity, while ALDH5A1 and COMP exhibited inverted U-shaped associations (both P-nonlinear < 0.05). For the transition from incident CVD to subsequent multimorbidity, one protein was inversely and eight proteins were positively associated with multimorbidity risk, with CD163 emerging as the top hit (HR per interquartile range: 1.37; 95% CI: 1.13, 1.66; $q = 0.012$). Among individuals with T2D, eight proteins were positively associated with multimorbidity, with CD209 being the top hit (HR: 1.43; 95% CI: 1.16, 1.76; $q = 0.005$) (Figs. 3 and 4). HR curves for linear and non-linear trends are visualized in Supplementary Fig. 3 for the proteomic signature scores and top hits for each transition to multimorbidity.

Among the proteins inversely related to MVPA, GGT1 was associated with risks of CVD and strongly with T2D as well as multimorbidity, illustrating context-dependent dose-response relations (Figs. 4 and Fig. 5A; Supplementary Table 3). These associations were non-linear positive with T2D risk, and linear positive with CVD risk and multimorbidity risk following a CVD diagnosis (e.g., multimorbidity, HR: 1.32; 95% CI: 1.12, 1.57; q = 0.015). Among the proteins positively associated with MVPA, we found seven that were associated with single diseases (CVD, T2D) as well as multimorbidity; and one protein (CHRDL2) that was associated with cancer, T2D, and multimorbidity (Figs. 4 and 5B; Supplementary Table 3). Across outcomes, some proteins showed consistent dose-response-like associations. For example, ALPP showed positive associations with both CVD and T2D (e.g., T2D, HR: 1.18; 95% CI: 1.10, 1.26; $q = 1.10 \times 10^{-5}$) and showed even stronger associations with multimorbidity (after CVD diagnosis, HR: 2.05; 95% CI: 1.34, 3.12; q = 0.025). However, some proteins demonstrated divergent patterns. For example, IGFBP1 was strongly inversely associated with T2D risk (HR: 0.52; 95% CI: 0.47, 0.56; $q = 1.20 \times 10^{-50}$) but was positively associated with CVD risk (HR: 1.08; 95% CI: 1.03, 1.14; q = 0.017) and with multimorbidity following T2D (HR: 1.37; 95% CI: 1.09, 1.73; q = 0.036).

### Sensitivity analyses

First, we assessed the association between the proteomic signature score and all outcomes after exclusion of MVPA from the models and observed minimal impact on model performance (Supplementary Table 4). Second mutual adjustment did not substantially alter the estimates for the proteomic signature score but attenuated the associations between MVPA and both T2D and multimorbidity following CVD (Supplementary Table 5). For example, the HRs for the proteomic signature score in relation to T2D risk remained virtually identical after mutual adjustment, whereas the corresponding MVPA association attenuated towards the null ($HR_{MVPA}$: 0.91; 95% CI: 0.86, 0.96; $HR_{mutual}$: 0.99; 95% CI: 0.93, 1.04). Third, after adjustment for ethnicity, LASSO retained 191 proteins (187 previously identified), with similar coefficient magnitudes as compared to the main signature (Supplementary Data 1). Fourth, including participants with an incident event during the first year of follow-up (n = 1279) yielded consistent results. Here, 858 proteins passed the pre-filtering step, and 140 were retained in the final signature. Of these, 132 overlapped with the 220 proteins identified in the main analysis, with concordant directions of effect (Pearson's r = 0.99

**A Case numbers per transition**

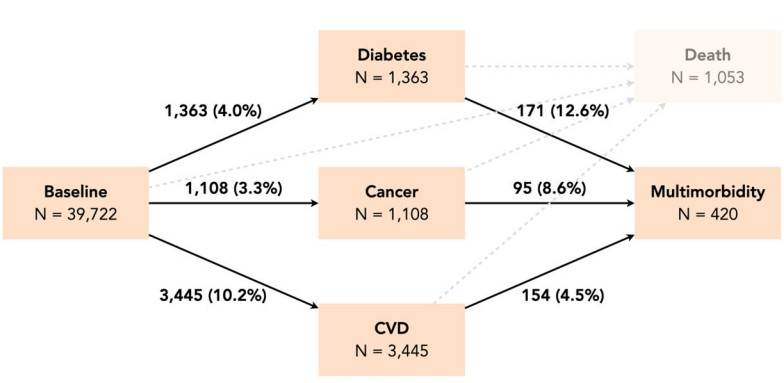

**B Protein correlations**

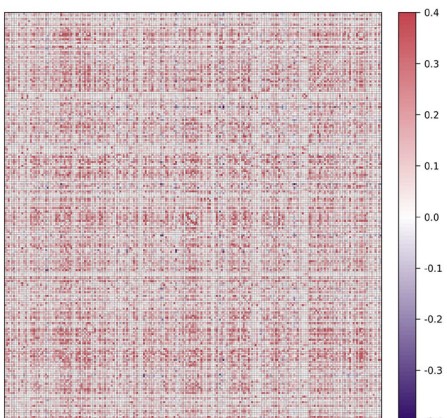

**C Physical activity related proteins**

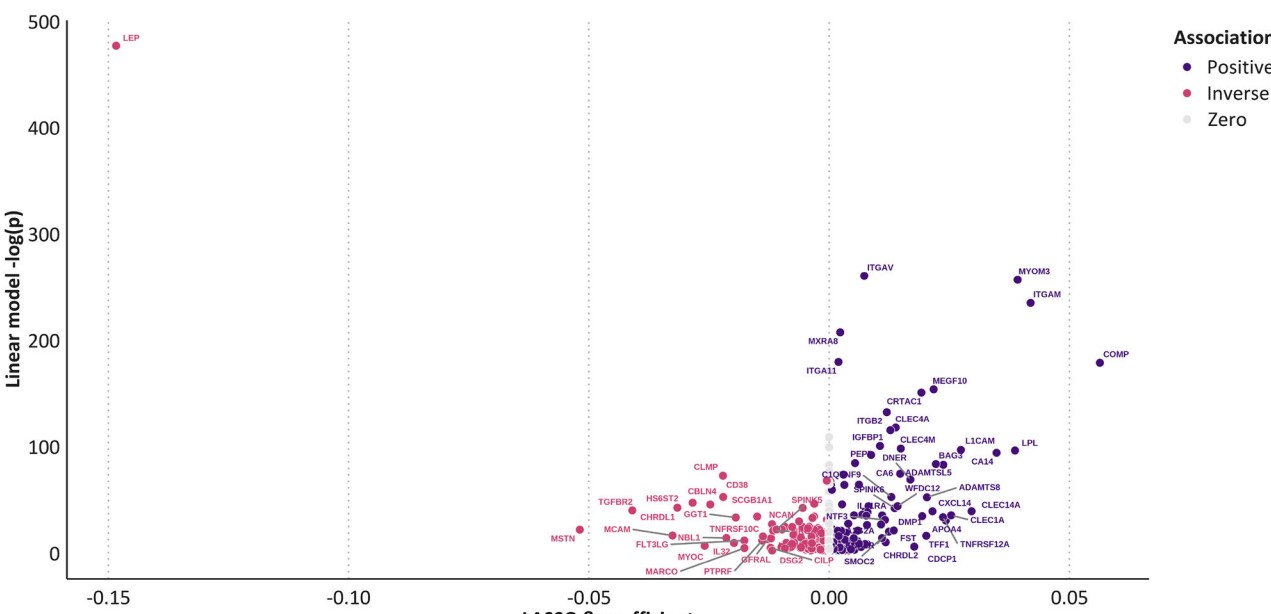

**Fig. 1 | Overview of disease transitions and protein analyses related to physical activity. A** Case numbers for each transition from disease-free to incident disease, from incident disease to subsequent multimorbidity, as well as death (for completeness). **B** Pairwise correlations for each physical activity-related protein. **C** Comparison of LASSO ß-coefficients (on the *x*-axis) and the negative logarithm of the *P*-value from linear regression models (on the y-axis) for the 220 identified proteins. *P*-values were obtained from two-sided likelihood ratio tests and adjusted for multiple testing using the false discovery rate method (Benjamini–Hochberg).

for beta coefficients), indicating little influence of latent disease on the observed associations. Last, of the 220 signature proteins, 176 had matching protein targets in EPIC. Using these overlapping proteins, the computed signature score was positively associated with recreational physical activity after adjustment for age, sex, and study center ($\beta = 3.2$ MET-minutes per 1-standard deviation increase; $p < 0.0001$; $R^2 = 0.13$).

## Discussion

We identified 220 proteins robustly associated with MVPA after correction for multiple testing. Our functional enrichment analysis revealed that many of these proteins likely play key roles in preserving tissue integrity, regulating immune and metabolic functions, and promoting tissue repair. We also found that 181 of the MVPA-related proteins were associated with at least one disease among cancer, CVD, and T2D, or progression to multimorbidity. Many of these proteins (n = 91) were associated with at least two

of these disease transitions, indicating potential shared biological pathways. These molecular associations are consistent with a beneficial role of physical activity in relation to disease burden.

Epidemiologic studies have found strong associations between physical activity and cancer and cardiometabolic diseases[8]. Some pathways whereby physical activity contributes to better health are well-established, including improvements in insulin sensitivity, reduction of chronic low-grade inflammation, and modulation of sex hormones[14]. In our study, we leveraged large-scale proteomic data to uncover additional, more specific molecular insights.

We found inverse associations with proteins involved in food intake, metabolism, as well as muscle and cell growth regulation, including LEP[49], MSTN[50], TGFBR2[51], and related proteins such as IGFBP3 and PTH. The inverse association between MVPA and pro-inflammatory LEP has been reported in other proteomic signatures of physical activity[21,22]. A previous

**A** Inversely associated proteins **B** Positively associated proteins

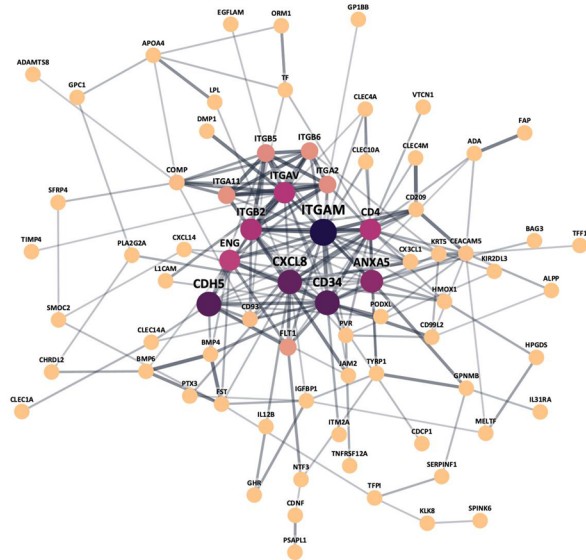

**Fig. 2 | Protein-protein interaction networks. A** Inversely with physical activity-related proteins. **B** Positively with physical activity-related proteins. The size of the node reflects the modularity level (MCC method), and the width of the edges reflects the interaction score.

**Table 2 | Pathway enrichment analysis for proteins inversely or positively related to physical activity**

| Category | Description | *q*-value |
|---|---|---|
| Inversely associated proteins | | |
| GO Cellular Component | Extracellular region | 0.0017 |
| Positively associated proteins | | |
| GO Cellular Component | Cell surface | <0.0001 |
| | Integrin complex | 0.0016 |
| | External side of plasma membrane | 0.0016 |
| | Integral component of membrane | 0.0021 |
| | Intrinsic component of membrane | 0.0021 |
| | Protein complex involved in cell adhesion | 0.0036 |
| | Cell periphery | 0.0197 |
| | Plasma membrane signaling receptor complex | 0.0103 |
| GO Molecular Function | Integrin binding | 0.0045 |
| | Opsonin binding | 0.0187 |
| | Protein-containing complex binding | 0.0330 |

*GO* Gene ontology.
No specific pathway showed significant enrichment in the Kyoto Encyclopedia of Genes and Genomes (KEGG), Reactome, or WikiPathways databases.

meta-analysis showed that physical activity may decrease LEP levels in individuals with disturbed glycemic control[52]. Obesity is strongly positively associated with LEP[53]. However, since body mass index was accounted for in the protein identification, physical activity is likely related to LEP independently of body fat.

MVPA was positively associated with levels of COMP, ITGAM, and MYOM3, which interact with other integrins (e.g., ITGAV, ITGB2, ITGA2, TGB5), highlighting involvement in immune cell adhesion, migration, and cartilage and muscle integrity[54,55]. COMP[56] and MYOM3[57] are known to be highly responsive to physical activity.

Pathway enrichment analysis indicated that proteins positively associated with MVPA were enriched for integrin complexes, receptor signaling, and opsonin binding, reflecting roles in immune recognition, tissue remodeling, and repair. Integrins facilitate immune cell migration to sites of inflammation and can influence tumor growth[58], while opsonins enhance pathogen clearance[59]. These findings collectively suggest that MVPA may

contribute to disease prevention by promoting immune resilience and maintaining tissue integrity.

We identified key proteins that may contribute to inverse associations between MVPA and cancer, CVD, and T2D. As previously described, we found positive relations between MVPA and integrins such as ITGA2, ITGB5, and ITGA11. These three proteins were inversely associated with cancer risk, potentially indicating a yet unknown mechanism between MVPA and proteins from the integrin complex, which generally functions as context-dependent tumor suppressors[58]. However, ITGA2 and ITGB5 are pivotal in promoting tumor cell growth and metastasis, and these proteins are often overexpressed in specific cancer tissues[60,61]. In addition, CLEC4A showed a strong inverse association with cancer risk. CLEC4A is an inhibitory receptor that plays a crucial role in regulating both innate and adaptive immune responses, particularly in the context of inflammatory conditions[62]. Nevertheless, there is currently no strong evidence connecting physical activity to the modulation of CLEC4A expression, but our results are supported by

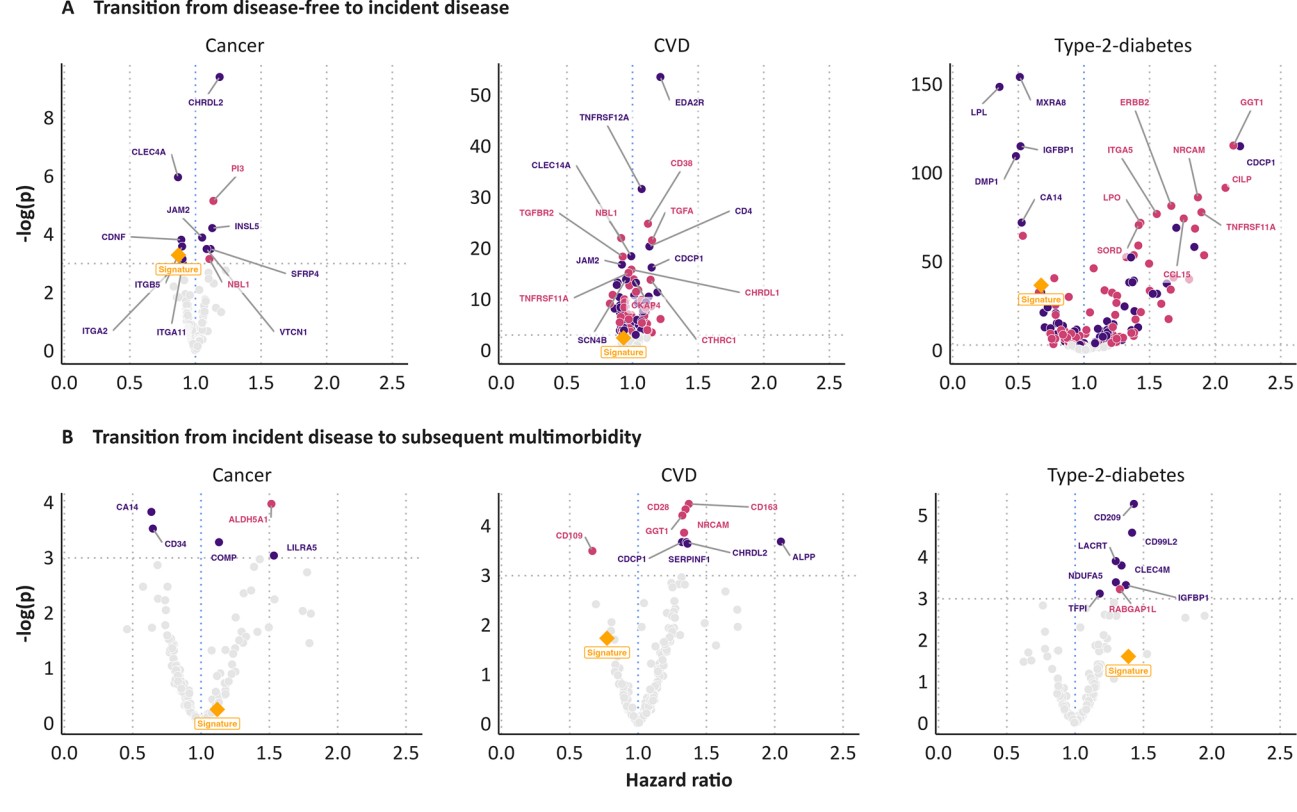

**Fig. 3 | Hazard ratios for all proteins and the proteomics signature for incident disease and multimorbidity. A** Transition from disease-free to incident disease. **B** Transition from incident disease to subsequent multimorbidity. *y*-axis shows the negative logarithm of the *P*-value. Cox proportional hazards models were used, two-sided, with *P*-values obtained from two-sided likelihood ratio tests and adjusted for multiple testing using the false discovery rate method (Benjamini-Hochberg). All models were stratified by age at baseline (5-year increments), sex, and country (England, Scotland, Wales), and adjusted for education level (highest, intermediate, lowest, none of those), socio-economic status (Townsend index, categorized using tertiles, missing values coded as missing), smoking (never, former, current), alcohol use (never, former, current), sedentary behavior (0–3 h, 4–5 h, 6–7 h, >8 h of daily TV watching, PC use during leisure, and driving), and screening for breast and/or bowel cancer (binary) as categorical variables, as well as physical activity (MET-hours), body mass index (kg/m²), and diet (healthy diet score, 0–7 scale) as continuous variables.

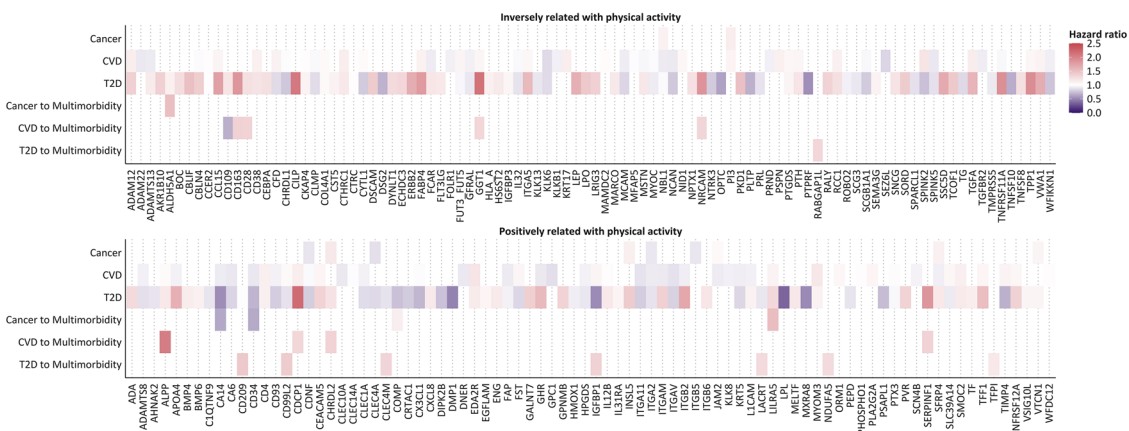

**Fig. 4 | Hazard ratios for related proteins after FDR-correction.** All models were stratified by age at baseline (5-year increments), sex, and country (England, Scotland, Wales), and adjusted for education level (highest, intermediate, lowest, none of those), socio-economic status (Townsend index, categorized using tertiles, missing values coded as missing), smoking (never, former, current), alcohol use (never, former, current), sedentary behavior (0–3 h, 4–5 h, 6–7 h, >8 h of daily TV watching, PC use during leisure, and driving), and screening for breast and/or bowel cancer (binary) as categorical variables, as well as physical activity (MET-hours), body mass index (kg/m²), and diet (healthy diet score, 0–7 scale) as continuous variables.

another recent UK Biobank study that found this protein associated with device-based and various self-reported measures of physical activity[23].

Inverse associations with T2D risk were observed for MXRA8, LPL, IGFBP1, and DMP1. Importantly, LPL, IGFBP1, and DMP1 are directly involved in metabolic regulation, impacting lipid and glucose metabolism and glucose homeostasis. Of these proteins, LPL demonstrates a particularly robust response to physical activity[63,64]. On the other hand, MXRA8, also referred to as DICAM, has been shown to enhance endothelial cell adhesion

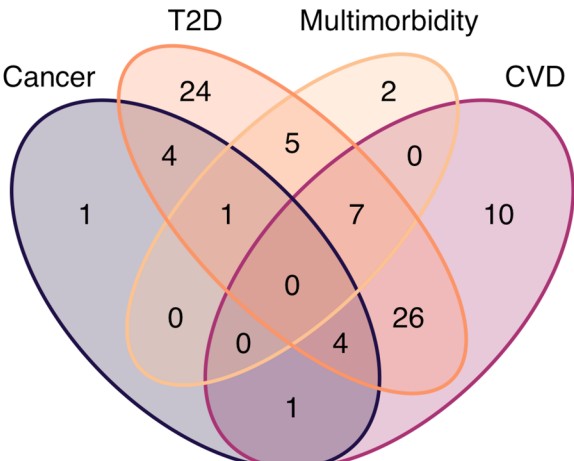

**Fig. 5 | Venn diagrams for proteins associated with physical activity by each outcome. A** Inversely with physical activity-associated proteins. **B** Positively with physical activity-associated proteins. CVD: cardiovascular disease; T2D: type 2 diabetes.

and migration, potentially functioning as a negative regulator of angiogenesis[65], and has been linked to decreased abdominal fat[66].

We found various proteins that were inversely associated with MVPA and positively associated with disease risk. Concerning CVD, half of the top hit proteins were inversely associated with MVPA, which were, in turn, positively associated with CVD risk. For example, CD38 and TGFA are involved in chronic inflammatory processes and endothelial function and play pivotal roles in blood pressure elevation and vascular dysfunction[67,68].

MVPA was strongly inversely correlated with LEP, which was associated with an increased risk of T2D. These findings support the notion of a physical activity-mediated pathway toward decreased T2D risk by lower circulating LEP levels[69,70]. Moreover, GGT1 was positively associated with T2D risk, consistent with meta-analytical evidence showing that higher serum levels of GGT1 contribute to increased T2D risk[71]. However, Mendelian randomization analyses have not substantiated the association[72]. CILP also showed an inverse association with MVPA but was generally positively associated with T2D. Proteins from the CILP family are predominantly expressed in cartilage cells but are also found in various other tissues, and elevated serum levels have been linked to impaired glucose metabolism[73,74].

Concerning cancer risk, PI3 (also referred to as Elafin) is typically expressed at low levels under normal physiological conditions, but it is highly upregulated in response to proinflammatory cytokines[75]. It has well-documented anti-inflammatory and antimicrobial properties through the inhibition of serine proteases involved in inflammation and bacterial defense[76]. In addition, PI3/Elafin expression is downregulated in cancer, while its overexpression inhibits cancer cell proliferation[77,78]. These mechanisms suggest that the observed positive association between PI3/Elafin and cancer – also reported in a previous UK Biobank study[79]—may reflect reverse causality driven by underlying inflammatory activity contributing to tumorigenesis, rather than a direct role of elevated PI3/Elafin in cancer development.

Findings for proteins associated with multimorbidity generally exhibited greater heterogeneity. For example, IGFBP1 (positively associated with MVPA) was inversely associated with T2D risk but positively associated with CVD and multimorbidity following T2D. Thus, IGFBP1 may be beneficial for metabolic health, as previously discussed, but potentially is adverse for cardiovascular health. This dual association has been noted in the literature, which suggests that low IGFBP1 levels indicate cardiometabolic risk, whereas higher levels are associated with worse cardiovascular outcomes[80,81]. Lastly, certain proteins, such as ALPP, demonstrated an

amplified risk for multimorbidity compared with individual disease risks, suggesting an elevated risk of additional conditions in comorbid individuals.

Our results indicate that some of the identified proteins may be associated with the progression from a first disease to multimorbidity, suggesting shared molecular patterns. However, certain proteins may exert opposing effects across disease transitions. Although proteins were measured in a disease-free state, their associations with first and second diseases may diverge, potentially due to dynamic changes in protein function following disease onset. For example, IGFBP1 may play different roles depending on the disease context, which could change after the first disease manifests. More generally, MVPA may upregulate levels of a given protein, making it protective against one disease in a healthy state. However, its role in other outcomes could shift once the first disease develops. Baseline protein levels reflect a general biological predisposition, but they cannot fully account for disease-induced alterations in protein function over time. Future studies are warranted to better understand these dynamic relationships. Additionally, for the transition to multimorbidity, collider bias may arise when conditioning on disease status (i.e., when selecting individuals based on having a first disease), potentially distorting associations between baseline protein levels and multimorbidity due to shared causes of both the exposure and the first disease. Such bias may weaken observed associations. In our study, risk estimates for some, but not all, proteins were in the expected direction and magnitude after conditioning on the first disease, suggesting that collider bias was of limited concern in these cases. In contrast, for proteins positively associated with physical activity, predominantly positive associations with the secondary incident diseases were observed; this pattern is likely influenced by collider bias. In addition, associations for multimorbidity were generally weaker and less consistent than for first disease risk, likely reflecting smaller numbers of cases or potential improvements in lifestyle behaviors, including MVPA after a first disease, which would attenuate associations with multimorbidity. These factors highlight that results for second incident diseases should be seen as exploratory and interpreted cautiously, underscoring the complexity of proteomic biomarkers across disease trajectories.

Paradoxically, some proteins exhibited divergent associations with MVPA and disease risk. For example, half of the top hit proteins for CVD were positively associated with both MVPA and CVD risk. These included tumor necrosis factor (TNF) family members EDA2R, TNFRSF12A, and TNFRSF10A, which have also been associated with increased CVD risk in previous studies from the UK Biobank and China Kadoorie Biobank[79,82]. In addition to its role as an inflammatory signaling TNF receptor, EDA2R –

also known as TNFRSF27 – functions as a marker of cancer-induced muscle atrophy and has been implicated in obesity-related glucose intolerance through inflammatory pathways involving c-Jun amino-terminal kinases (JNKs), which are activated via EDA2R[83,84]. The top hit for cancer risk, CHRDL2, exhibited both increased cancer risk and a positive relation with MVPA, and is a known oncogene in gastrointestinal cancers[85,86].

These divergent findings highlight the complexity of proteome-wide association studies and the need for cautious interpretation. The health impact of elevated protein levels depends on whether they reflect adaptive and compensatory responses to physical activity (e.g., immune activation or tissue remodeling), or pathological activation that promotes disease. For instance, proteins involved in inflammatory, immune, or tissue stress pathways can be mobilized by MVPA as part of an adaptive response, but in some individuals may also signal subclinical disease activity, e.g., cardiovascular stress such as exercise-induced cardiac biomarker release or maladaptive cardiac remodeling[87]. Higher proteomic signature scores for MVPA were associated with reduced risks of cancer, CVD, and particularly T2D, suggesting that MVPA-related proteomic profiles may reflect underlying biological states associated with lower disease risk. The strongest associations were observed for T2D, with clusters of proteins positively associated with MVPA being inversely related to T2D risk, and vice versa, contributing to a clear inverse association between the proteomic signature and T2D. The proteomic signature score was not associated with the development of multimorbidity. However, sensitivity analysis showed that also MVPA alone was not associated with multimorbidity, supported by prior findings[11], suggesting that neither MVPA nor its proteomic signature at baseline may be associated with a subsequent disease once a chronic condition is present. Follow-up assessments of physical activity would be required to account for potential changes of physical activity after diagnosis of a first disease. To explore potential mediation, we compared models with and without mutual adjustment for the proteomic signature score and MVPA. The signature score remained associated with disease outcomes after adjustment for MVPA, whereas MVPA-disease associations attenuated after adjustment for the signature score, suggesting that the signature captures MVPA-related proteomic variation relevant to disease risk. We note that proteomic signature scores may serve as proxies for an exposure (here, MVPA) by capturing complex, multivariate patterns. While they can enhance exposure classification, their composite nature limits their utility for elucidating biological mechanisms, which requires analysis of individual proteins with known or hypothesized roles in relevant pathways.

Our findings suggest potential proteomic pathways through which physical activity may impact multimorbidity. Given that physical activity is a modifiable behavior, these results can inform intervention studies. Future studies that examine causality (e.g., through genetic approaches and/or causal mediation analysis) for the proteins identified in our work would be welcome.

Our study has several strengths. First, we examined the relationship between proteins, MVPA, and disease risk in an unprecedentedly large study encompassing 2911 proteins. Second, to minimize potential reverse causation affecting protein levels, we excluded prevalent cases along with incident cases occurring within the first year of follow-up. Third, we identified MVPA-related proteins using a rigorous approach that included comprehensive quality assessment of protein measurements, a two-step feature selection process to mitigate collider bias, and nested cross-validation to assess model performance. Fourth, cross-platform validation in an independent cohort supported the robustness of the identified proteomic signature as well as its generalizability at least to Caucasian ancestries. However, our study also had limitations. We relied on self-reported MVPA from a validated questionnaire, whose measurements are prone to exposure misclassification, potentially attenuating true associations between physical activity and circulating proteins. Conversely, the proteomic signature score of MVPA may reduce classical random reporting error through aggregation across multiple proteins (analogous to polygenic risk scores). This is supported by our mutual adjustment analysis, where the association of self-reported MVPA with, for example, T2D, attenuated towards the null

while the signature score remained unaffected. However, the proteomic score is not error-free and likely reflects additional biological variability unrelated to MVPA. Accordingly, the proteomic score should be treated as a complementary exposure capturing a biological correlate of MVPA rather than an error-free substitute for questionnaire-based MVPA. Furthermore, our findings primarily reflect habitual engagement in higher-intensity activity and may not capture the full spectrum of physical activity. Additionally, analyzing individual proteins likely oversimplified their complex interactions, potentially underestimating their collective role in disease pathways. We addressed this concern by conducting functional enrichment and interaction network analyses. Some of the diverging patterns of associations between protein–physical activity and protein–outcome relationships may be the result of common limitations of observational research, prone to residual confounding and reverse causality. Both physical activity and proteomic measurements were obtained at baseline only; thus, temporal changes in behavior or protein expression could not be captured, limiting our ability to assess dynamic relationships with chronic disease development. Lastly, the study participants are of predominantly European ancestry, limiting the generalizability of the findings to other populations.

## Ethics approval and consent to participate
Ethical approval was obtained from the North West Multi-Center Research Ethics Committee. All participants provided written informed consent. This study has been conducted in line with the Declaration of Helsinki.

## Conclusion
Our findings suggest that physical activity-induced modulation of proteomic pathways involved in cell adhesion, immune surveillance, and extracellular remodeling may contribute to enhanced tissue integrity and reduced disease risk. These findings provide insights into biological pathways and molecular correlates associated with physical activity and health.

## Data availability
The data that support the findings of this study are available from the UK Biobank, which is an open-access resource. Bona fide researchers can apply to use the UK Biobank dataset by registering and applying at http://ukbiobank.ac.uk/register-apply/. Source data for the Figs. 1B, 1C, 3, and 4 are provided with the supplementary information file 'Supplementary Data 2' of this paper.

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

## Acknowledgements

Funding for IIG_FULL_2021_027 was obtained from the World Cancer Research Fund (WCRF UK), as part of the World Cancer Research Fund International grant program. This study was supported by the French National Cancer Institute (l'Institut National du Cancer, INCA_16824), the German Research Foundation (BA 5459/2-1). Support for this work was funded in part by the European Union's Horizon Europe research and innovation program (under the Marie Skłodowska-Curie Doctoral Network ColoMARK; Grant Agreement No. 101072448 to DW). The UK Biobank was supported by the Wellcome Trust, Medical Research Council, Department of Health, Scottish government, and Northwest Regional Development Agency. It has also had funding from the Welsh Assembly government and the British Heart Foundation. The research was designed, conducted, analyzed, and interpreted by the authors entirely independently of these funding sources. The funder had no role in study design, data acquisition and analysis, decision to publish, or preparation of the manuscript. The coordination of EPIC-Europe is financially supported by the International Agency for Research on Cancer (IARC) and also by the Department of Epidemiology and Biostatistics, School of Public Health, Imperial College London, which has additional infrastructure support provided by the NIHR Imperial Biomedical Research Center (BRC). The national cohorts are supported by Associazione Italiana per la Ricerca sul Cancro-AIRC-Italy, Italian Ministry of Health, Italian Ministry of University and Research (MUR), Compagnia di San

Paolo (Italy); Dutch Ministry of Public Health, Welfare and Sports (VWS), the Netherlands Organisation for Health Research and Development (ZonMW), World Cancer Research Fund (WCRF), (The Netherlands); Instituto de Salud Carlos III (ISCIII), Regional Governments of Andalucía, Asturias, Basque Country, Murcia and Navarra, and the Catalan Institute of Oncology-ICO (Spain); Cancer Research UK (C864/A14136 to EPIC-Norfolk; C8221/A29017 to EPIC-Oxford), Medical Research Council (MR/N003284/1, MC-UU_12015/1 and MC_UU_00006/1 to EPIC-Norfolk; MR/Y013662/1 to EPIC-Oxford) (United Kingdom). Previous support has come from the "Europe against Cancer" Program of the European Commission (DG SANCO). The generation of the proteomic data was partly funded by the Michael J Fox Foundation (#008994 to Christina M. Lill and Elio Riboli), the Cure Alzheimer's Fund (to Christina M. Lill and Lars Bertram), the "CReATe-Clinical Research in ALS and Related Disorders for Therapeutic Development" Consortium (to Christina M. Lill and Lars Bertram), with additional grant support from the Heisenberg program of the Deutsche Forschungsgemeinschaft (DFG; LI 2654/4-1 to Christina M. Lill). SomaScan® data were generated under Master Research Agreement, 14th December 2021, between Imperial College London and SomaLogic Inc. SomaLogic was not involved in analyzing or interpreting the data, or in writing or submitting the manuscript for publication. We thank the participants of the European Prospective Investigation into Cancer and Nutrition (EPIC) study for their valuable contribution to this research. We acknowledge the use of data from: EPIC-Milan cohort, principal investigator Sabina Sieri; EPIC-Florence: PI Giovanna Masala; EPIC-Turin: PI Carlotta Sacerdote; EPIC-Ragusa: PI Rosario Tumino; EPIC-Cambridge cohort, PI Nick Wareham; and EPIC-Oxford cohort, PI Ruth Travis. We thank the National Institute for Public Health and the Environment (RIVM), Bilthoven, Netherlands, for their contribution and ongoing support to the EPIC study. This research has been conducted using the UK Biobank Resource under Application Number 55870, and we express our gratitude to the participants and those involved in building the resource. This work uses data provided by patients and collected by the NHS as part of their care and support; therefore, for the data linkage, we want to acknowledge NHS England (Copyright © (2023), NHS England. Re-used with the permission of the NHS England [and/or UK Biobank]. All rights reserved). This research used data assets made available by National Safe Haven as part of the Data and Connectivity National Core Study, led by Health Data Research UK in partnership with the Office for National Statistics and funded by UK Research and Innovation (research which commenced between 1st October 2020–31st March 2021 grant ref MC_PC_20029; 1st April 2021–30th September 2022 grant ref MC_PC_20058). Where authors are identified as personnel of the International Agency for Research on Cancer/ World Health Organization, the authors alone are responsible for the views expressed in this article, and they do not necessarily represent the decisions, policy or views of the International Agency for Research on Cancer/ World Health Organization.

## Author contributions

M.J.St and H.F. had full access to all the data in the study and took responsibility for the integrity of the data and the accuracy of the data analysis. Study design: M.J.St, V.V., and H.F.; Acquisition, analysis, or interpretation of the data: M.J.St, H.B., P.B., R.C., P.F., B.F., C.M.F., M.J.G., L.P.N., D.W., C.O.M., M.J.Sá, M.D.C., M.F.L., V.V., and H.F.; Manuscript writing: M.J.St, M.F.L., and H.F.; Critical revision of the manuscript for important intellectual content: H.B., P.B., R.C., P.F., B.F., C.M.F., M.J.G., L.P.N., D.W., M.F.L., V.V., and H.F.

## Funding

## Competing interests

The authors declare no competing interests.

## Consent for publication

Not applicable.

## Additional information

[1]Department of Epidemiology and Preventive Medicine, University of Regensburg, Franz-Josef-Strauß-Allee 11, Regensburg, Germany. [2]Department of Epidemiology and Preventive Medicine, Medical Sociology, University of Regensburg, Dr.-Gessler-Str. 17, Regensburg, Germany. [3]Faculty of Life Sciences, Department of Nutritional Sciences, University of Vienna, Vienna, Austria. [4]International Agency for Research on Cancer (IARC), Nutrition and Metabolism Branch, 25 Avenue Tony Garnier, CS90627, Lyon, France. [5]Department of Prevention Cancer Environment, Centre Léon Bérard, Lyon, France. [6]INSERM UMR1296 Radiation: Defense, Health, Environment, Lyon, France. [7]Department of Cancer Epidemiology and Prevention Research, Cancer Care Alberta, Alberta Health Services, Calgary, AB, Canada. [8]Departments of Oncology and Community Health Sciences, Cumming School of Medicine, University of Calgary, Calgary, AB, Canada. [9]Cancer Epidemiology and Prevention Research Unit, School of Public Health, Imperial College, London, UK. [10]Julius Center for Health Sciences and Primary Care, University Medical Center Utrecht, Utrecht University, Utrecht, The Netherlands. [11]Escuela Andaluza de Salud Pública, Granada, Spain. [12]Instituto de Investigación Biosanitaria de Granada, Granada, Spain. [13]Centro de Investigación Biomédica en Red de Epidemiología y Salud Pública, Madrid, Spain. [14]Department of Epidemiology, Regional Health Council, IMIB-Arrixaca, Murcia University, Murcia, Spain. ✉e-mail: michael.stein@helmholtz-munich.de

