## [Transparent Peer Review file · Communications Medicine]

Proteomics signature of moderate-to-vigorous physical activity and risk of multimorbidity of cancer and cardiometabolic diseases

Corresponding Author: Dr Michael Stein

Version 0:

Reviewer comments:

Reviewer #1

(Remarks to the Author)

The study examined proteomics, physical activity, and multimorbidity (cancer, CVD and T2D) in the UK Biobank. The overall finding is that several physical activity related proteins were associated with incident diseases and multimorbidity. This is a straightforward study. However, the study would benefit from clearer justification, additional analyses, and more cut-and-dry discussion.

Specific comments:

1. P3. Para 1. Given the heterogeneity of cancer (different organs) and CVD (in heart, brain, or vessels), how convincing is it to say, "shared etiological pathways"?
2. P3. Para 3. The assumption here is that proteome is stable to reflect "long-term" physical activity. However, there are various aspects of physical activity, including strength, habit, and timing. If both acute and long-term physical activity influence proteome, how useful and complex would this tool be?
3. P3. Para 4. It appears that the variability of associations between proteome and physical activity is high. Did authors have comments on it?
4. P4. Para 2. If the authors focused on moderate-to-vigorous physical activity, it should be justified in the introduction, and the title should be modified, as mentioned above, physical activity is not a single trait.
5. P4. Para 4. Better specify "exposure" and "protein identification".
6. P5. Para 1. Okay to focus on "moderate-to-vigorous physical activity", which cannot represent "physical activity". I suggest modifying throughout the manuscript. Also, this should be notified in the limitations.
7. P5. Para 3. No surgical codes for CVD?
8. P5. Para 3. For multimorbidity, how did authors calculate their follow-up time? From first disease to the incidence of the second one, or from the beginning of follow-up? Actually, participants cannot develop multimorbidity if they did not have the first disease.
9. P6. Para 2. Race/ethnicity was not adjusted. Did authors have justification for it?
10. P6. Para 2. I assume "pre-identified proteins" in step 2 were identified in step 1. What's the identification criteria, p value or q value?
11. P6. Para 2. Similarly, what's the criteria for "the most predictive proteins"?
12. P6. Para 2 and 3. Authors identify physical activity related proteins and calculate the proteomic signature score both in the UK Biobank, which would intuitively lead to significant findings. Have authors considered external validation? For example, in the Introduction section, the authors mentioned other studies, like in FHS. This is important.
13. P6. Para 3. Why choose non-zero coefficients rather than proteins with significant associations?
14. P6. Para 4. The correlation between the score and original moderate-to-vigorous physical activity is pretty weak, which may indicate inappropriate methods.
15. If the basic assumption holds, the paper will benefit from mediation analyses, showing proportion of incidence of disease mediated by proteomic signature.
16. Results. Suggest adding reference significance line for every volcano plot. Or other ways to show significance.
17. P10. Para 3. "Strongly" or "Significantly"?
18. The paper will benefit from less dense discussion about pathway analyses results, and additional discussion about potential implications of findings. The discussion would also benefit from mediation analyses due to potential causal clues.
19. P15. Para 2. Authors have options to test generalizability, for example, using associations information in FHS to

calculate the score.

Reviewer #2

(Remarks to the Author)

This manuscript presents a comprehensive analysis using proteomic data from 33,806 participants in the UK Biobank to investigate associations between moderate-to-vigorous physical activity and the risk of cancer, cardiovascular disease (CVD), type 2 diabetes (T2D), and multimorbidity. Using a two-stage modeling approach—linear regression followed by LASSO regression—the authors derive a proteomic signature for physical activity and evaluate its associations with incident disease and transitions to multimorbidity. The manuscript is well-organized, methodologically robust, and offers novel insights into the biological mechanisms potentially linking physical activity and chronic disease risk. However, several clarifications and refinements would improve the precision and interpretability of the findings.

Major Comments

1. Multimorbidity is defined as having any two or more of the three outcomes (cancer, CVD, T2D). However, it would strengthen the analysis to distinguish individuals with one, two, or all three diseases. This stratification would allow readers to assess whether individual proteins or the overall proteomic signature is differentially associated with increasing disease burden and whether a dose–response pattern exists.
3. While excluding cases in the first year of follow-up helps reduce reverse causality, subclinical disease may still influence baseline proteomic profiles. It would be helpful to know whether associations differ when early cases are included versus excluded. In addition, the authors could briefly note that incorporating causal frameworks such as Mendelian randomization to help identify proteins that may have a direct role in disease etiology.
3. The authors report using fivefold nested cross-validation to derive the proteomic activity score, showing a correlation coefficient ($r = 0.32$) with physical activity. It would be helpful to clarify whether correlation is the most appropriate metric to evaluate the predictive performance or representativeness of the score. Given the goal of creating a surrogate biomarker, additional performance metrics (e.g., R^2 , calibration, discrimination) might be more informative.
4. Please clarify whether both proteomic and physical activity data were collected only at baseline. If no follow-up measurements were used, this should be acknowledged as a limitation, since changes in behavior or protein expression over time could influence disease risk. Single-timepoint exposure and biomarker data may limit the ability to capture dynamic relationships with chronic disease development.

Minor Comments

5. For proteins discussed in the main text (e.g., GGT1, CLEC4A), consider including a brief functional description or a list of full names in a table or supplementary material. This will improve accessibility for readers less familiar with proteomic nomenclature.
6. The description of spline modeling is appropriate, but please indicate whether proportional hazards assumptions were tested in the Cox models. Also clarify how non-linearity was assessed, as this will improve transparency in reporting.
7. The terms “transition from baseline to disease” and “transition from disease to multimorbidity” may be unclear or overly general. Consider rephrasing to “transition from disease-free to incident disease” and “transition from incident disease to subsequent multimorbidity,” which better reflect temporal sequencing and event structure.

Reviewer #3

(Remarks to the Author)

This study represents a valuable contribution to the understanding of the biological mechanisms linking physical activity (PA) to chronic disease risk, particularly focusing on cancer, cardiovascular disease (CVD), type 2 diabetes (T2D), and their multimorbidity. The strengths include a large, well-characterized cohort from the UK Biobank, comprehensive adjustment for potential confounders, and rigorous statistical methods. The integration of proteomic data with prospective outcome data provides novel insights into the molecular pathways through which PA may influence disease risk.

Major Concerns

1. The study uses the International Physical Activity Questionnaire (IPAQ) to measure PA, which is subjective and prone to recall bias, social desirability bias, and misclassification. Device-based measures (e.g., accelerometers) would provide more objective and precise estimates of PA intensity and duration. The authors acknowledge this limitation but do not discuss how misclassification might affect the validity of the proteomic signature or the strength of associations with outcomes. For example, underestimation of PA could attenuate the observed relationships between PA, protein levels, and disease risk.
2. Several proteins positively associated with PA are also linked to increased disease risk (e.g., CHRD2 with cancer, EDA2R with CVD). The discussion acknowledges the complexity of these relationships but does not adequately explore potential mechanisms.

Version 1:

Reviewer comments:

Reviewer #1

(Remarks to the Author)

The authors mostly met my comments.

Reviewer #2

(Remarks to the Author)

I acknowledge the revisions made to the manuscript and the responses to my previous questions.

I recommend the manuscript for publication.

Reviewer #3

(Remarks to the Author)

Thank you for the opportunity to re-review the revised manuscript. While the authors have made considerable edits, several fundamental methodological flaws and logical gaps remain. These issues critically undermine the validity and reliability of the study's conclusions. Therefore, I cannot recommend this manuscript for publication in its current form. The major concerns are as follows:

- 1.The validity of the "MVPA proteomic signature" is questionable due to the use of self-reported physical activity data. The study's foundation is a proteomic signature derived from the IPAQ questionnaire, a tool notoriously prone to recall and social desirability bias. Consequently, the 220-protein signature may not reflect the biology of actual MVPA, but rather the characteristics of individuals who report high levels of activity. This population may differ systematically in diet, lifestyle, and health consciousness, introducing uncontrolled confounding that likely drives the observed protein associations.
- 2.The study makes unjustified claims of causality and mechanistic insight. As an observational study, it cannot establish the "biological mechanisms" repeatedly claimed in the Abstract and Discussion. The findings are susceptible to reverse causation, where baseline protein levels or subclinical disease states could influence an individual's ability to exercise, rather than the other way around.
- 3.The analysis of multimorbidity yields inconsistent and unconvincing results. The proteomic signature score was only associated with multimorbidity risk when CVD was the first incident disease, but not when cancer or T2D occurred first (lines 458-459). This inconsistency substantially weakens the central hypothesis that physical activity influences these diverse diseases through a common proteomic pathway.
- 4.The multimorbidity analysis is at high risk of collider bias. By restricting the analysis to individuals who have already developed a primary disease, the study conditions on a collider. This can create spurious associations between baseline proteins and the risk of a second disease. While the authors briefly acknowledge this risk (line 427), this major methodological flaw is not adequately addressed, calling the validity of these results into question.
- 5.The 220-protein signature score is a statistical "black box" with limited biological interpretability. The score was generated using LASSO regression, a machine-learning method optimized for prediction, not for identifying causal biological factors. The resulting score is a statistical construct that lacks clear biological meaning.
- 6.The use of a single composite score offers little granular scientific insight. Condensing the complex information from 220 different proteins into a single score, while statistically convenient, obscures the individual roles of distinct proteins and pathways. This approach prevents a deeper understanding of how specific biological processes may link physical activity to disease.

Version 2:

Reviewer comments:

Reviewer #3

(Remarks to the Author)

Thank you for the revisions. I have no further comments and recommend acceptance.
